

# Intergenerational implications of alcohol intake: metabolic disorders in alcohol-naïve rat offspring

Pawel Mierzejewski, Alicja Zakrzewska, Julita Kuczyńska, Edyta Wyszogrodzka and Monika Dominiak

Department of Pharmacology, Institute of Psychiatry and Neurology, Warsaw, Poland

## ABSTRACT

Alcohol drinking may be associated with an increased risk of various metabolic diseases. Rat lines selectively bred for alcohol preference and alcohol avoidance constitute an interesting model to study inherited factors related to alcohol drinking and metabolic disorders. The aim of the present study was to compare the levels of selected laboratory biomarkers of metabolic disorders in blood samples from naïve offspring of Warsaw alcohol high-preferring (WHP), Warsaw alcohol low-preferring (WLP), and wild Wistar rats. Blood samples were collected from 3-month old (300–350 g) alcohol-naïve, male offspring of WHP ($n = 8$) and WLP rats ($n = 8$), as well as alcohol-naïve, male, wild Wistar rats. Markers of metabolic, hepatic, and pancreatic disorders were analysed (levels of homocysteine, glucose, total cholesterol, triglycerides and $\gamma$-glutamyl transferase (GGT), aspartate (AST), alanine aminotransferase (ALT), and amylase serum activities). Alcohol-naïve offspring of WHP, WLP, and wild Wistar rats differed significantly in the levels of triglycerides, total cholesterol, homocysteine, as well as in the activity of GGT, ALT, AST, and amylase enzymes. Most markers in the alcohol-naïve offspring of WHP rats were altered even thought they were never exposed to alcohol pre- or postnatally. This may suggest that parental alcohol abuse can have a detrimental influence on offspring vulnerability to metabolic disorders.

## INTRODUCTION

Heavy alcohol drinking can lead to serious health problems, including cardiovascular diseases (CVD) and liver cirrhosis (*Triglyceride Coronary Disease Genetics Consortium and Emerging Risk Factors Collaboration et al., 2004*; *Ronksley et al., 2011*; *Yang et al., 2008*). In numerous studies, heavy alcohol drinking was linked to both an increased risk of CVD and metabolic syndrome, leading to the hypothesis that alcohol-induced metabolic disorders could stand behind the relationship between excessive alcohol consumption and CVD (*Athyros et al., 2007*; *Alkerwi et al., 2009*; *Kahl & Hillemacher, 2016*). The above-mentioned associations may be related to environmental as well as genetic factors. Twin, family, and adoption studies have consistently shown that genetic factors play an important role in the development of alcohol dependence (*Kendler et al., 1992*; *Kendler, Davis & Kessler, 1997*; *Heath et al., 1997*). Genetic loading is also an important risk factor for

Corresponding author
Monika Dominiak, mdominia@wp.pl

dyslipidaemia, diabetes, and metabolic syndrome (*Anjos & Polychronakos, 2004*; *Joy et al., 2008*; *Blakemore & Buxton, 2004*). Hence, one may hypothesise that excessive alcohol consumption and metabolic abnormalities are related to the same genetic mechanism(s).

Prenatal alcohol consumption has been shown to interfere with the development of multiple organs and systems, resulting in Foetal Alcohol Spectrum Disorders (FASD) (*Jones, 2011*). Epidemiological studies have revealed that alcohol is the most common teratogen to which humans are exposed. It was estimated that eight children per 1,000 live births are diagnosed with some degree of FASD (*Lange et al., 2017*). This condition is characterised by a spectrum of structural defects, central nervous system disorders, as well as growth deficits, and metabolic disorders (*Roozen et al., 2016*; *Carter et al., 2016*). Although it is perceived as a childhood disorder, the majority of health problems, including growth restriction, hyperinsulinemia, and other metabolic disruptions are lifelong, persisting into adulthood (*Carter et al., 2013*; *Moore & Riley, 2015*).

Apart from this well-known unfavourable effect on the foetus, recent studies on rats have highlighted the possible negative impact on offspring of maternal alcohol exposure also around the time of conception. Several lines of evidence have suggested that periconceptional alcohol use may have a negative impact on adult offspring, including increased obesity risk, altered plasma lipids, leptin profiles, and liver steatosis (*Gardejber et al., 2018*). Furthermore, the study of *Dorey et al. (2018)* revealed a greater risk of obesity in the offspring of females exposed to alcohol around the time of conception. It was associated with the altered expression of key components of the mesolimbic reward pathway and an increased preference for palatable foods in offspring.

However, apart from reports on maternal alcohol use during pregnancy or in the periconceptional period, studies on paternal preconception alcohol exposure on offspring also provide interesting findings. Paternal alcohol intake during the preconception period induces a spectrum of health problems and alters gene expression in rodent offspring (*Knezovich & Ramsay, 2012*; *Finegersh et al., 2015*; *Jabbar et al., 2016*; *Rompala & Homanics, 2019*). The results of the study of *Kim et al. (2014)* suggest that the preconceptional exposure to ethanol of male mice before mating induces ADHD-like behaviours in offspring, possibly via epigenetic changes in gene expression. *Asimes et al. (2017)* demonstrated that adolescent binge ethanol exposure altered DNA methylation patterns in the hypothalamus of alcohol-naïve offspring. In another study examining parental binge alcohol abuse, differences in the expression of genes involved in neurogenesis, reproductive function, and regulation of obesity were observed in offspring (*Przybycien-Szymanska et al., 2014*). Also, the studies of *Chang et al. (2019a)* and *Chang et al. (2019b)* using a mouse model of paternal alcohol exposure have identified the markers of hepatic fibrosis and abnormalities in both lipid production and insulin signaling in the offspring of alcohol-exposed sires. The aforementioned studies raise the possibility that alcohol affects naïve offspring even in the absence of direct foetal alcohol exposure, and epigenetic inheritance may be the mechanism responsible for this phenomenon.

In light of the metabolic changes associated with alcohol consumption, whether common metabolic markers differ in alcohol-naïve offspring of high- and low-preferring rats is of interest. Rat lines selectively bred for alcohol preference and alcohol avoidance constitute

an interesting model to study the inherited factors related to alcohol drinking and metabolic disorders. The breeding of rats was carried out as previously described by *Rok-Bujko, Dyr & Kostowski (2006)*. Specifically, Warsaw high-preferring (WHP) and Warsaw low-preferring (WLP) rats were bred from Wistar rats for opposite ethanol preference (*Rok-Bujko, Dyr & Kostowski, 2006*; *Dyr & Kostowski, 2008*). When given a choice between water and 10% ethanol, WHP rats voluntarily drink excessive amounts of ethanol (>5.0 g/kg/24 h), while WLP rats drink marginal quantities of ethanol (*Dyr & Kostowski, 2008*).

The aim of the present study was to compare the common markers reflecting cardiovascular, liver, and pancreatic function in alcohol-naïve offspring of WHP, WLP, and Wistar rats: the levels of homocysteine, glucose, total cholesterol, triglycerides, $\gamma$-glutamyl transferase (GGT), aspartate aminotransferase (AST), alanine aminotransferase (ALT), and amylase serum activities. We hypothesise that markers of cardiovascular, liver, and pancreatic function differ between alcohol-naïve offspring of WHP, WLP, and Wistar rats.

## MATERIAL AND METHODS

### Rats and experimental design

Twenty-four (eight per group), twelve-week-old, alcohol-naïve Warsaw high-preferring (WHP), alcohol-naïve Warsaw low-preferring (WLP) rats, and male albino Wistar rats (Charles River, Sulzfeld, Germany) were used for the study. The WLP and WHP rats were obtained from the Department of Pharmacology and Physiology of the Nervous System at the Institute of Psychiatry and Neurology in Warsaw, Poland.

During the selection, which is a standardized procedure repeated in every generation, WHP and WLP rats were exposed to alcohol for 30 days, after reaching a weight of 250 g. In males, this occurs at around 60 days of age. When given a choice between water and 10% ethanol, WHP rats voluntarily drank excessive amounts of ethanol (>5.0 g/kg/24 h), while WLP rats consumed negligible quantities of ethanol (*Dyr & Kostowski, 2008*). Periods of binge drinking are observed only in WHP rats. In this study, all parents of examined offspring were previously subject to a standardized selection procedure. After a 30-day period of alcohol exposure, they were mated 24 h after the last ethanol consumption and subsequent offspring were produced. The parents were not intoxicated at the time of mating and were not exposed to alcohol at any time during pregnancy, thus, the analysed offspring were never directly exposed to alcohol. Two rats per litter were used (4 litters in total for each group).

All the rats were housed in the same environmental conditions and handled by the same experimenters for at least 30 days before blood sampling. The environmental conditions were as previously described in the study of *Zakrzewska et al. (2020)*. Specifically, the rats were housed in standard plastic cages, in groups of three, with wood chip bedding on the floor, in fully controlled environmental conditions : temperature 22 $\pm$ 2 °C, humidity 50% $\pm$5, and 12-hour light/12-hour dark cycles with lights on at 8.00 a.m. All the rats had unlimited access to tap water and standard laboratory chow (Labofeed H, WPIK, Kcynia, Poland) throughout the duration of the study.

The offspring of all rat lines were weighed at 21, 60, and 90 days of age. There were no differences in body weight between the tested rat lines at any stage of their development

or their food intake (both $F < 1.0$, $p > 0.05$). In the last measurement at 90 days of age (at the time of sampling) the offspring of WHP, WLP, and Wistar rats weighed 345 g ($\pm 27.6$), 338 g ($\pm 30.5$) and 341 g ($\pm 25.8$), respectively.

Blood samples were obtained from 3-month-old (average of 300-350 g) drug- and alcohol-naïve male offspring of WHP ($n = 8$) and WLP rats ($n = 8$). Non-selected 3-month-old, drug- and alcohol-naïve male Wistar rats (Charles River, Sulzfeld, Germany; 300-350 g; $n = 8$) were used as additional controls. The rats were deprived of food for 12 h overnight and were sacrificed by rapid decapitation with a guillotine designed for rodents. Trunk blood was collected into a tube immediately after decapitation. Serum was obtained after centrifugation at 1600 g for 15 min. The serum was collected into a plain tube and stored at $-80$ °C until the analysis. No haemolysis was observed in any collected sample.

The Ethics Committee for Animal Care provided its consent for the study (Agreement no. 10/2010). The study was performed in full compliance with respective Polish and European ethical regulations (Directive No. 86/609/EEC).

## Biochemical analysis

The activities of AST, ALT, GGT, and amylase, as well as the levels of glucose, total cholesterol, and triglycerides were measured in the serum using automated enzymatic colorimetric methods (Cobas Integra, Roche Diagnostics, Rotkreuz, Switzerland). Serum homocysteine determination was carried out using fully automated chemiluminescence assay (ADVIA Centaur, Siemens Healthcare Diagnostics, Erlangen, Germany). The assays were performed in accordance with the manufacturers' instructions in the analytical laboratory of the Institute of Psychiatry and Neurology in Warsaw.

## Statistical analysis

A one-way ANOVA was used to evaluate between-group differences in the biochemical measures. The Newman-Keuls test was applied for post hoc comparisons. The normality of distribution of the tested variables was verified using the Shapiro–Wilk test. For two tested variables: ALT and triglycerides, the distribution could not be considered as normal. For the analysis of these parameters, the Kruskal-Wallis test was applied, while Dunn's multiple comparisons test was used for post hoc comparisons. $P$-values of less than 0.05 were considered significant. The study was exploratory in nature, thus, no correction for multiple comparisons was applied. The Statistica 12.0 software package (StatSoft, Inc., Tulsa, OK, USA) was used to analyse all the data.

## RESULTS

The one-way ANOVA revealed that the offspring of WHP, WLP, and Wistar rats differed significantly in the following laboratory parameters: homocysteine ($F_{(2,21)} = 9.65$, $p = 0.001$), total cholesterol ($F_{(2,21)} = 44.63$, $p < 0.001$), AST ($F_{(2,21)} = 6.74$, $p = 0.005$), amylase ($F_{(2,21)} = 14.30$, $p < 0.001$), and GGT ($F_{(2,21)} = 135.87$, $p < 0.001$) (Table 1). There were no differences in blood glucose levels ($F_{(2,21)} = 0.86$, $p = 0.437$).

The post hoc Newman-Keuls comparisons indicated that the alcohol-naïve offspring of WHP rats had significantly higher levels of total cholesterol and homocysteine, as well as
**Table 1  Biochemical characteristics of the alcohol-naïve offspring of WHP, WLP and alcohol-naïve wild Wistar rats.**

| Parameters | WHP (n = 8) | WLP (n = 8) | Wistar (n = 8) | P-value |
|---|---|---|---|---|
| Homocysteine (μmol/L) | 7.48 (1.37)[a,b] | 4.77 (1.55) | 6.57 (0.68) | [a] $p < 0.001$; [b] $p = 0.009$ |
| Glucose (mg/dL) | 143.5 (13.55) | 149.83 (12.11) | 143.16 (8.03) | All $p > 0.05$ |
| Total cholesterol (mg/dL) | 79.37 (16.30)[a,b] | 34.87 (3.87)[c] | 46.00 (2.72) | [a] $p < 0.001$; [b] $p < 0.001$; [c] $p = 0.033$ |
| Triglycerides (mg/dL) | 74.28 (22.94)[a,b] | 25.06 (4.63) | 37.91 (2.47) | [a] $p < 0.001$; [b] $p < 0.001$ |
| ALT (U/L) | 97.75 (13.18)[a,b] | 74.25 (7.95)[c] | 63.81 (5.18) | [a] $p = 0.04$; [b] $p < 0.001$; [c] $p = 0.02$ |
| AST (U/L) | 200.5 (22.78)[b] | 177.66 (29.34) | 152.53 (25.87) | [b] $p = 0.004$ |
| GGT (U/L) | 4.95 (0.74)[a,b] | 9.85 (0.54) | 9.67 (0.72) | [a] $p < 0.001$; [b] $p < 0.001$ |
| Amylase (U/L) | 2174.37 (359.72)[b] | 2073.25 (189.26)[c] | 1559.37 (131.57) | [b] $p < 0.001$; [c] $p < 0.001$ |

Notes.
[a] Offspring of WHP rats vs offspring of WLP rats.
[b] Offspring of WHP rats vs Wistar rats.
[c] Offspring of WLP rats vs Wistar rats.
Values represent mean (SD); p-value from Newman-Keuls test or Dunn's test (for ALT and triglycerides).

significantly higher AST and amylase activity as compared to the alcohol-naïve WLP and the Wistar controls. The offspring of WHP rats had a significantly higher level of homocysteine as compared to the offspring of WLP rats, but not the Wistar rats. Surprisingly, the offspring of WHP subjects showed significantly lower GGT activity as compared to the WLP and the Wistar rats.

Regarding ALT as well as triglycerides, the Kruskal-Wallis test revealed a highly significant difference between the tested groups: $H(2) = 17.6$, $p < 0.001$, $H(2) = 20.2$, $p < 0.001$, respectively. Post hoc comparisons for these markers revealed significant differences between WHP and Wistar rats (both $p < 0.001$) for both markers, as well as between WHP and WLP rats ($p = 0.04$, $p \leq 0.001$, for ALT and triglycerides, respectively).

## DISCUSSION

This is the first study to show changes in selected biochemical markers of metabolic disorders in naïve Wistar rats selected for high alcohol preference (WHP rats). A major finding of the present study is that the alcohol-naïve offspring of WHP rats have significantly higher blood levels of total cholesterol, triglycerides, and homocysteine than their WLP counterparts and wild Wistar rats. Some differences were also observed in the markers of hepatic and pancreatic function between alcohol-naïve offspring of WHP, WLP, and Wistar rats (ALT, AST, and amylase were increased, whereas GGT activity was decreased in WHPs).

Interestingly, our previous study showed an increased level of leptin—a risk factor for obesity and hypertension—in alcohol-naïve WHP rats as compared to alcohol-naïve WLP rats (*Mikołajczak et al., 2002*). Although leptin is involved in lipid metabolism (*Hynes & Jones, 2001*), the exact mechanism of the above observation in alcohol-naïve WHP rats remains unclear. In other studies, chronic alcohol exposure in rodents was shown to reduce the serum level of leptin (*Otaka et al., 2007*). Furthermore, leptin deficiency was associated with the development of fatty liver in mice (*Tan et al., 2012*).

A high level of total cholesterol or triglycerides is associated with an increased risk of atherosclerosis, myocardial infarction, and stroke (*Lindenstrøm, Boysen & Nyboe, 1994*; *Langsted et al., 2010*; *Sarwar et al., 2010*). Elevated levels of homocysteine are thought to facilitate the development of atherosclerosis and are considered a risk factor for coronary disease and stroke (*Lawrence de Koning et al., 2003*; *Guthikonda & Haynes, 2006*). Several studies have shown a positive correlation between the amount of alcohol consumption and the blood homocysteine level (*Van der Gaag et al., 2000*; *Gibson et al., 2008*). It has also been shown that active drinking and early abstinent alcoholics have a higher level of homocysteine in the serum as compared to the general population (*Bleich et al., 2000*; *Bleich et al., 2005*). There are also some epidemiological analyses showing a link between familial alcoholism and obesity among men and women (*Grucza et al., 2010*).

There are many hypotheses aimed at explaining the increased risk of metabolic diseases in alcoholics. Our study indicates that some inherited factors (genetic, epigenetic) may play a role. These factors transmitted from alcoholic parents to their offspring could, at least partially, be responsible for the increased risk of metabolic disorders. A few studies have attempted to identify the genes responsible for the differences in alcohol preference among the naïve offspring of high- and low-preferring rats (*Carr et al., 2007*; *Kimpel et al., 2007*; *Stankiewicz et al., 2015*). One such study indicated that alcohol preference may share some genetic factors with the cardiovascular system and metabolism (*Stankiewicz et al., 2015*), however, this study failed to identify any gene differences between high- and low-preferring rats. Thus, it does not support the hypothesis that the observed phenotypic differences are gene related. One can hypothesise that other factors, for example, the epigenetic mechanism, may have a dominant role in the observed phenomenon. Many reports in the literature have indicated that environmental factors can have long-term effects on gene expression through epigenetic mechanisms (*Shukla et al., 2008*). Epigenetics refers to mechanisms that modify gene expression without any changes in the DNA sequence. Recent evidence suggests that epigenetic modifications can be carried to the offspring via the germline and are responsible for the transmission of alcohol-related disorders across generations (*Mahnke, Miranda & Homanics, 2017*; *Chastain & Sarkar, 2017*). Interestingly, there are studies showing the association of parental alcohol intake with subsequent changes in gene expression in naïve offspring (*Rompala & Homanics, 2019*; *Kim et al., 2014*; *Asimes et al., 2017*), as well as with changes in lipid metabolism (*Przybycien-Szymanska et al., 2014*). Moreover, *Pennington, Shuvaeva & Pennington (2002)* showed that maternal preconceptional alcohol intake may be associated with hypertriglyceridemia in adult offspring who have never been exposed to alcohol. Paternally inherited alterations in epigenetic programming were also shown to be related to metabolic defects observed in foetal alcohol spectrum disorders. In rodents, chronic paternal alcohol use affected insulin signaling and lipid homeostasis in the offspring through paternally inherited alterations in liver × receptor activity (*Chang et al., 2019a*). Similar findings were reported by *Chang et al. (2019b)*, assessing long-term impacts of chronic preconception paternal alcohol use in mice. Evidence of metabolic programming in offspring included suppressed cytokine profiles within the liver and pancreas, consistent with findings from this study.

In our study, most of the tested hepatic biomarkers in the alcohol-naïve offspring of WHP rats had changed similarly to the heavy alcohol abusers. Surprisingly, the serum activity of the most specific alcohol-related biomarker, GGT, was decreased as compared to the WLP counterparts and the Wistar controls. These contrary results are difficult to explain. GGT, unlike AST and ALT, is located on the external surface of cellular membranes. Chronic alcohol use is considered to induce membrane GGT solubilization and increase the release of the enzyme, thus, the mechanism of release of given liver enzymes is related to the degree of cell damage. One can also hypothesise that different epigenetic mechanisms regulate the activity of these enzymes (*Teschke, 2018*) and one of the possible mechanisms could be linked to the changes in cellular membrane properties. Nevertheless, the mechanism responsible for this is yet to be discovered.

The results of this study should be interpreted in light of certain limitations. WHP and WLP line breeding involves the process of selecting rats in each generation through exposure to alcohol. Therefore, the design of the study could not include a control group of the offspring of WHP unexposed rats and, similarly, for this reason distinguishing between the paternal and maternal effects was also unfeasible as both males and females were required to be exposed to alcohol. Also, due to ethical concerns, we were only able to include a limited number of rats per group, thus, only male offspring were examined. However, there are sex-specific effects on offspring growth and long-term metabolic programming (*Chang et al., 2019b*), thus, comparable studies on female offspring are warranted.

It should be emphasized that although the offspring were not directly exposed to alcohol, the last estrogenic cycle for females, as well as spermatogenesis in males, were exposed to alcohol. Thus, it cannot be excluded that periconceptional exposure could have an impact on gametes and affect, for example, oocyte maturation (*Fleming et al., 2004*; *Bielawski et al., 2002*; *Kalisch-Smith et al., 2019*). Similarly, this may influence the perinatal behaviours of parents and food intake in such a way that it may have indirect effects on metabolism. It has been shown, for example, that exposure to alcohol three weeks before conception affects the birth weight and body growth of offspring (*Jabbar et al., 2016*). In our study, however, we failed to observe any differences in weight gain between the examined rat lines.

## CONCLUSIONS

In conclusion, the present study revealed that the alcohol-naïve offspring of high alcohol preference rats differed significantly from their low alcohol preference counterparts and wild Wistar rats in terms of common markers reflecting cardiovascular, liver, and pancreatic disorders. To our knowledge, this is the first study showing the differences in selected biochemical markers of metabolic disorders in high- and low-preferring rats. One can hypothesise that these changes are associated with epigenetic changes at the level of the gametes and/or with the genetically inherited phenotype. Either way, this study suggests that parental alcohol abuse can have a detrimental influence on offspring vulnerability to metabolic disorders.
Given the detrimental influence of excessive alcohol consumption and metabolic disorders on individuals and societies, the present findings should be verified in further preclinical and clinical studies.

### Funding
This work was supported by the Institute of Psychiatry and Neurology (grant no. 501-38-004-18023). The funders had no role in study design, data collection and analysis, decision to publish, or preparation of the manuscript.

### Grant Disclosures
The following grant information was disclosed by the authors:
Institute of Psychiatry and Neurology: 501-38-004-18023.

### Competing Interests
The authors declare there are no competing interests.

### Author Contributions
- Pawel Mierzejewski and Monika Dominiak conceived and designed the experiments, analyzed the data, prepared figures and/or tables, authored or reviewed drafts of the paper, and approved the final draft.
- Alicja Zakrzewska and Edyta Wyszogrodzka performed the experiments, prepared figures and/or tables, and approved the final draft.
- Julita Kuczyńska performed the experiments, prepared figures and/or tables, authored or reviewed drafts of the paper, and approved the final draft.

### Animal Ethics
The following information was supplied relating to ethical approvals (i.e., approving body and any reference numbers):

The study was performed in full compliance with ethical standards laid down in respective Polish and European regulations (Directive No. 86/609/EEC). The local Committee for Animal Care approved of all the experimental procedures (Agreement No. 10/2010).

### Data Availability
The raw data are available in the Supplemental File.

### Supplemental Information
Supplemental information for this article can be found online at http://dx.doi.org/10.7717/peerj.9886#supplemental-information.

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
