# Peer review of "Intergenerational implications of alcohol intake: metabolic disorders in alcohol-naïve rat offspring"

_PeerJ, doi:10.7717/peerj.9886_

## Round 0.1 · original submission · Major Revisions

As noted by the reviewers, to improve this study, there are a number of issues that need to be addressed, especially in the experimental design and in the methodology.

Reviewer 1 ·

Basic reporting

The paper is written clearly and well referenced.

Experimental design

The research question is well defined, however, the experiments are very limited in scope. No mechanisms of epigenetic inheritance are assayed nor do the authors perform any pathophysiological examinations of the offspring liver. The altered measures of hepatic function are compelling.

From the materials and methods, it is unclear how long the male rats (paternal) were exposed to alcohol prior to mating.

What % alcohol did the paternal rats drink?

When were the body weights of the male offspring examined? Do the authors have any data prior to weaning?

How old were the male offspring when they were terminated? (Were these three-month-old animals as well?)

Why were only the male offspring examined? (why not include females as well?)

Validity of the findings

Although the scope of this study is limited, the data clearly show an impact of paternal drinking on base measures of hepatic function within the male offspring. As can be seen above, I have just a few technical comments and questions for the authors.

Additional comments

The questions posed in section 2 above need to be included in the materials and methods.

Reviewer 2 ·

Basic reporting

This study proposes that consumption of alcohol can result in transgenerational programming of metabolic disorders in offspring. The authors tested this by examining systemic biomarkers for metabolic disorders in alcohol naïve offspring from rat lines selectively bred for alcohol preference and compared this with wild-type Wistar rats. Although this is an interesting approach, there are some flaws in the design and presentation/analysis of data that make it difficult to follow through to the conclusions drawn by the authors.

Raw data was shared and the article followed the appropriate structure. Sufficient field background/context was provided and appropriate, recent references were cited. However, there was repetition of data from figures in the table (see comment below).

Grammatical/writing style/spelling errors: Some examples where the language could be improved or there were typos include lines 52, 119, 192, 202, 227 but there are also other examples. Please review the text.

Experimental design

Lines 38-39: The abstract states that “Most markers in the alcohol-naïve offspring of WHP rats were altered similarly to heavy alcohol abusers”. However, there was no control group of offspring that were not alcohol-naïve as they describe to draw this conclusion. Can the authors please comment on why they did not include such a group that would allow a direct comparison of measured outcomes in their study?

What is the justification for only including male offspring in the study? There are well-known sex-specific differences in programming of metabolic outcomes by different maternal insults so this needs to be at least acknowledged in the discussion.

Line 121 Methods: 8 male rat offspring from each line were used in the study, but it is unclear how many litters these came from. How many offspring were used per litter? Anything >1-2 per litter will introduce litter effects. Also, although the overall weight range for offspring was provided, what was the mean +/- SD weight of offspring within each group?

Statistics: Were data sets tested for a normal distribution before using the parametric one-way ANOVA? This should be done and stated in the methods. If not normally distributed, a non-parametric Kruskal-Wallis test should be used. Given the difference in variance in the WHP group in the triglyceride data, Welch’s correction (or something equivalent) needs to be applied. Similarly for the ALT data.

Validity of the findings

Although the authors state that the offspring were not directly exposed to alcohol, ethanol exposure only ceased 24h prior to mating in both males and females. This means that the last estrous cycle for females was exposed and so the final stages of oocyte growth and maturation were exposed to alcohol. Similarly, spermatogenesis was exposed to alcohol in males. 5g/kg is a high dose of EtOH for the WHP rats and could directly impact on these gametes. The authors very briefly mention this as a limitation of the study at the end of the discussion, but it is a fundamental point that impacts on whether this is a genetically inherited phenotype or purely the result of epigenetic changes to the germline in male and female breeders used for this study. Can the authors please comment further on this?

Figures: As submitted, these are not of sufficient quality for publication. The labels on the X and Y axes are unclear, as are the axes labels. Graphs within each figure should be designated as A) and B) and referred to in the figure legend. It is convention to arrange the data with the control group, in this case the Wistar rats, to the left of the graph and then treatments of increasing severity to the right. Therefore, these figures should have data columns rearranged as Wistar, WLP, WHP. Figures should graph mean +/- SD (or SEM) rather than 95% confidence interval and provide the n numbers for each group. The box to represent the mean is not required, as the column graph already does this. The Y-axis should have the origin at 0. The p values for the significant differences shown in the Figures needs to be indicated (i.e. different symbols for P<0.05, P<0.01, P<0.001 etc).

Table: Data for glucose, ALT, triglycerides and GGT should be removed as they are already presented in Figures 1 and 2.

Supp file: Raw data should show units for all measures and define all abbreviations for clarity. Why is glucose and ASPAT data for the WHP group not shown to 1 decimal place, as per the other groups?

Additional comments

There is not enough data presented here to warrant a full publication. Perhaps a brief report?

·

Basic reporting

The manuscript is well written, in clear English. There are only a few minor typographical and grammatical errors.

The introduction provides context. I think it would be useful to include a brief description of the scale of the problem for humans. For example, are there appropriate references that could be cited in terms of the number of children with developmental issues resulting from Foetal Alcohol Spectrum Disorders?

In the table, the alignment of the p-values for the first two rows makes these a little unclear.

For both figures, the text on the axes (including axis titles) is narrow and very difficult to read. A different font should be used. In addition, the symbols to denote statistically significant results are too small to be easily seen and interpreted.

In general, the results are clear and relate to the stated aims. However, I believe the authors could have included a clear hypothesis.

Experimental design

The purpose of the studies is clear, and the authors describe the importance of understanding effects of alcohol abuse. As noted already, a clear hypothesis would be helpful.

In the introduction, the authors state that (line 71): “Paternal alcohol intake during the preconception period induces a spectrum of health problems and alters gene expression in rodent offspring.” However, in their experimental design (line 111): “[all parents].. were mated 24 hours after the last ethanol consumption and subsequent offspring were produced. The parents were not intoxicated at the time of mating.” Does this mean that both male and female parents consumed alcohol prior to mating? In that case, presumably the authors were not attempting to distinguish between paternal and maternal effects. I think this should be made clear, because in the introduction both maternal and paternal effects are described. Is there a reason why either maternal or paternal effects were not investigated, rather than having both?

Methods

Animal study:
The animal studies generally appear to have been carried out carefully and appropriately. For example, it is reassuring (line 115) that the rats were handled by the same experimenters for at least 30 days before blood sampling.

There are a few items I think should be addressed:

Line 101: On what basis did the authors choose 8 animals per group? Had some form of power analysis been undertaken? If so, what data was this based on? If not, why not? Did the authors have prior data to suggest 8 was sufficient?

Methods, line 127: Centrifugation is reported as 4500 rpm. Please report this as x g.

The authors describe analysis of both serum and plasma, but they only report on collection of serum. Were both serum and plasma collected, or were all analyses only done using serum?

In general, the statistical methods seem appropriate. My question is with the use of the Newman-Keuls test; one of the assumptions for this test is that the observations are independent. Can the authors comment on whether, for example, AST and ALT levels can be considered independent? If not, would another post-hoc analysis be more appropriate in this case?

Validity of the findings

My main issue with the discussion is that, in the introduction and discussion, it is noted that there are potential epigenetic mechanisms relating to effects of parental alcohol consumption on outcomes in offspring, and that seems to be an important factor in the observed results. The authors also note that these may be a subject for further studies. Some explanation of why no genetic or epigenetic outcomes were assessed in the current study would be helpful. Given these were identified as possibly important, it would have been helpful to include them. Do the authors plan to investigate such mechanisms?

Lines 181 to 183, the authors mention increased leptin levels in a previous study, but don’t link this to their current study. How might this be related to the current results?

The authors comment that the unexpected change in GGT might be due to different epigenetic mechanisms (line 223). Other than the single reference they cite, are there any other studies to support this, or which describe other mechanisms which may be relevant? If so, these should be included.

Additional comments

No further comments.

---

## Round 0.2 · Major Revisions

There are still some aspects that need to be addressed in the basic reporting, experimental design, and statistical analysis, as well as speculations about the interpretation of the results. The English language should be re-checked and improved.

Reviewer 2 ·

Basic reporting

- Although figure 1 has been altered, it still essentially repeats the data provided in Table 1 and, as such, does not add additional information. Therefore, only either Figure 1, with data presented as mean and SD, or Table 1 should be presented. What does the % on the Y-axis for Figure 1 represent? This is presumably as a % of the control group? This is unclear.
- There are still issues with grammar/writing style/spelling errors: Some examples where the language could be improved include lines 115, 166, 202-203, 266, 285 but there are also other examples. Please review the text.

Experimental design

- It needs to be made clearer in the methods how many litters the experimental animals came from. The authors addressed this in their response (i.e. 2 rats per litter were used, therefore 4 litters total for each treatment to provide n=8 per group) but it needs to be explicitly stated.
- It is appreciated that the authors have updated their description of the analysis to include mention of testing for normality and adjustment to a non-parametric test where appropriate. However, use of the Mann-Whitney U test as a post-hoc test between groups following the Kruskal Wallis test is not an appropriate test, as it does not adjust for multiple testing. A test such as Dunn’s multiple comparison procedure is required. This is the equivalent of using the Student Newman-Keuls test following a parametric one-way ANOVA.

Validity of the findings

It has been made clearer that this is a preliminary study, and the authors appear well aware of any limitations because of this and directions for future research. I agree with a comment made by one of the other reviewers that their preliminary results are compelling, but may be more reflective of epigenetic changes at the level of the gametes, rather than evidence for a genetically inherited phenotype.

Additional comments

The authors have adequately addressed most of the reviewers’ comments, but some issues with data presentation/analysis remain. In my opinion, this manuscript does not provide enough data to warrant a full publication, but perhaps a brief research report is appropriate.

---

## Round 0.3 · Minor Revisions

The authors have satisfactorily addressed the reviewers' comments. However, there are some issues concerning the language that need to be fixed, as indicated by reviewer 2.

Reviewer 2 ·

Basic reporting

The authors have adequately addressed all previous comments. However, there are still some minor errors in grammar/writing style as detailed below (NB: line numbers refer to the tracked changes version of the manuscript when viewed in ‘All Markup’ mode):
line 90-92. Rewrite to: “In light of the metabolic changes associated with alcohol consumption, whether common metabolic markers differ in alcohol-naïve offspring of high- and low-preferring rats is of interest.”
line 104, 280: remove “the”
line 117-119. Rewrite to: “During the selection, which is a standardized procedure repeated in every generation, WHP and WLP rats were exposed to alcohol for 30 days, after reaching a weight of 250 g. In males, this occurs at around 60 days of age.”
line 120: should be “drank”
line 126: rewrite to: “…never directly exposed to alcohol.”
line 198: rewrite to “This is the first study to show changes in selected biochemical markers of metabolic disorders in naïve Wistar rats selected for high alcohol preference (WHP rats).”
line 229: should be “phenotypic”
line 243: “…paternally inherited alterations in liver x receptor activity.”
line 244-247. Rewrite to: “Similar findings were reported by Chang et al. (2019b), assessing long-term impacts of chronic preconception paternal alcohol use in mice. Evidence of metabolic programming in offspring included suppressed cytokine profiles within the liver and pancreas, consistent with findings from this study.”
line 250: “…specific alcohol-related biomarker, GGT, was decreased…”
line 258: remove “the” before light.
line 264: replace “allowed” with able.
line 268. rewrite to: “…the last estrogenic cycle for females, as well as spermatogenesis in males, were exposed…”

Experimental design

No comment

Validity of the findings

No comment

Additional comments

Additionally, remove the sentence beginning “Based on this study…” at lines 282-284. Rewrite following sentence to “One can hypothesise that these changes are associated with epigenetic changes at the level of the gametes and/or with the genetically inherited phenotype. Either way, this study suggests that parental alcohol abuse can have a detrimental influence on offspring vulnerability to metabolic disorders.”

---

## Round 0.4 · accepted · Accept

The authors have satisfactorily addressed all the issues raised.